

# Anatomy of the 2018 agricultural drought in The Netherlands using in situ soil moisture and satellite vegetation indices

Joost Buitink[1], Anne M. Swank[1], Martine van der Ploeg[1, 2], Naomi E. Smith[3], Harm-Jan F. Benninga[4], Frank van der Bolt[5, 6], Coleen D. U. Carranza[2], Gerbrand Koren[3], Rogier van der Velde[4], and Adriaan J. Teuling[1]

[1]Hydrology and Quantitative Water Management Group, Wageningen University & Research, Wageningen, the Netherlands
[2]Soil Physics and Land Management Group, Wageningen University & Research, Wageningen, the Netherlands
[3]Meteorology and Air Quality Group, Wageningen University & Research, Wageningen, the Netherlands
[4]Department of Water Resources, Faculty of Geo-Information Science and Earth Observation (ITC), University of Twente, Enschede, the Netherlands
[5]Water Authority Aa en Maas, 's Hertogenbosch, the Netherlands
[6]Wageningen Environmental Research, Wageningen University & Research, Wageningen, the Netherlands

**Correspondence:** Adriaan J. Teuling (ryan.teuling@wur.nl)

**Abstract.** The soil moisture status near the land surface is a key determinant of vegetation productivity. The critical soil moisture content determines the transition from an energy-limited to a water-limited evapotranspiration regime. This study quantifies the critical soil moisture content by comparison of in situ soil moisture profile measurements of the Raam and Twenthe networks in the Netherlands, with two satellite derived vegetation indices (NIRv and VOD) during the 2018 summer

drought. The critical soil moisture content is obtained through a piece-wise linear correlation of the NIRv and VOD anomalies with soil moisture on different depths of the profile. This nonlinear relation reflects the observation that negative soil moisture anomalies develop weeks before the first reduction in vegetation indices. Furthermore, the inferred critical soil moisture content was found to increase with observation depth and this relationship is shown to be linear and distinctive per area, reflecting the tendency of roots to take up water from deeper layers when drought progresses. The relations of non-stressed towards water-

stressed vegetation conditions on distinct depths are derived using Remote Sensing, enabling the parameterization of reduced evapotranspiration and its effect on GPP in models to study the impact of a drought on the carbon cycle.

## 1 Introduction

Droughts can have wide environmental and socio-economic impacts, ranging from their effects on climate, the carbon cycle, food security, to water availability. Droughts are typically induced by a lack of precipitation and/or an above-average atmo-

spheric demand for evapotranspiration (ET), which leads to an associated reduced availability of soil moisture in the root zone (Seneviratne et al., 2010; Teuling, 2018). The former is typically referred to as meteorological drought, whereas the latter is referred to as agricultural drought. On the one hand, reduced soil moisture limits the plant water uptake and ET, leading to a shift in the land surface energy balance towards an increase in sensible heat flux compared to latent, hence establishing a positive feedback by exacerbating temperature and vapor pressure deficit increases through land-atmospheric feedbacks (Seneviratne





et al., 2010; Miralles et al., 2019; Lansu et al., 2020). On the other hand, reduction in ET through the closing of plants' stomata also affects the carbon cycle by reducing Gross Primary Production (GPP) (van der Molen et al., 2011; Reichstein et al., 2013). This can turn ecosystems from carbon sinks to sources, such as during the 2003 European summer drought and heatwave where GPP was reduced by as much as 30% (Ciais et al., 2005). While meteorological droughts are generally well-understood since they can be monitored by routine meteorological observations, quantifying the links between soil moisture, ET and vegetation

during agricultural drought is more challenging. This is the aim of the current study, where we focus on the record breaking drought of 2018 in Europe (Bakke et al., 2020).

Typically, two evapotranspiration regimes are distinguished: an energy limited regime where ET is highly sensitive to changes in available energy, and a water-limited regime where ET is highly sensitive to soil moisture conditions. As a result, the corresponding relation in the water limited regime between soil moisture and ET is often conceptualized and parameterized

by a bilinear function, separating the two regimes at the so-called critical soil moisture content (Seneviratne et al., 2010). There is considerable evidence that the strong nonlinearity is typical for most regions and conditions. This makes it key to i) predict the onset of drought impact on ET, and ii) predict the timescale of ET decay during drought (Teuling et al., 2006; Boese et al., 2019). In early field experiments, it was already observed that the actual ET fell below the potential only at lower levels of soil moisture, and that the value at which this occurred depended on the rate of potential ET (Denmead and Shaw, 1962). In

more recent studies at larger scales, it has been observed that ET rates over the summer increased rather than decreased in parts of Central-Western Europe during drought (Teuling et al., 2013), and that vegetation productivity in Alpine regions also increased during the 2003 summer drought (Jolly et al., 2005). In a recent study on vegetation-soil moisture coupling using satellite observation products (Denissen et al., 2020), it was found that the critical soil moisture is located at the lower rather than higher part of the soil moisture range. However, the remote sensing products used in these studies are subject to significant

limitations, mainly caused by the limited penetration depth of the microwave sensing used to estimate soil moisture compared to depths over which vegetation may take up water, and as a result the critical soil moisture content is still highly uncertain.

The impact of drought has been studied extensively using ecosystem level information obtained from eddy covariance sensors (i.e. FLUXNET), satellite-derived observations (van der Molen et al., 2011), or terrestrial biosphere modelling (van Schaik et al., 2018). This has provided valuable insight into the timing and impact of drought on vegetation productivity, as

quantified by GPP (Sippel et al., 2018; Stocker et al., 2019). Frankenberg et al. (2011), for example, showed that the spatio-temporal patterns of GPP are correlated with Solar-Induced chlorophyll Fluorescence (SIF), a satellite product which measures the re-emission of light by chloroplasts during photosynthesis. Koren et al. (2018) found that GPP decreases during droughts and can be quantified from reduction in SIF. This is confirmed other studies (e.g. Li et al., 2018) where SIF showed similar dynamics as GPP obtained by FLUXNET sites, and responds to stress due to low soil moisture (Jiao et al., 2019). Badgley

et al. (2017, 2019) found that SIF correlates strongly with satellite obtained Near-Infrared Reflectance of terrestrial vegetation (NIRv) and proposes to use this as proxy for GPP. Vegetation optical depth (VOD) is another satellite obtained measure to determine above ground vegetation water content using microwave sensors (Konings et al., 2016; Moesinger et al., 2020). VOD is proven to represent plant productivity (Teubner et al., 2018, 2019). Both NIRv and VOD have the advantage of a high





temporal resolution, in contrast to SIF data. This allows for a more precise analysis on how plant productivity is related to soil
moisture.

Even though a few FLUXNET sites provide in situ soil moisture measurements, these do not represent the entire root zone
or soil moisture dynamics at the eddy covariance flux footprint scale due to the large spatial variability of soil moisture even at
small scales (Teuling et al., 2013). As a result, many studies on drought response at FLUXNET sites or using satellite-derived
observations rely on calculated soil moisture proxies (e.g. Granier et al., 2007; Boese et al., 2019). Such proxies do not provide
a direct or quantitative insight into the propagation of soil moisture anomalies in the root zone or critical soil moisture content.

Whereas ecosystem flux observations and satellite observations of vegetation can provide valuable insight into the ecosystem
response to drought, they do not provide direct insight into processes that occur below the surface, in particular the timing,
location, and strategy of plant water uptake in the root zone. The parameterization of root water uptake during drought is thus
a major source of uncertainty in models (Braud et al., 2005; Teuling et al., 2006; Kumar et al., 2015; Combe et al., 2016). It
is well known that plants take up water from the upper soil layers first, and can compensate for a developing lack of moisture
available near the surface by increasing their uptake deeper in the profile to values much higher that can be expected based on
the root density (Sharp and Davies, 1985; Green and Clothier, 1995). Currently, many studies rely on the use of surface soil
moisture to diagnose drought processes. This is problematic, because surface soil moisture that can be measured by satellite-
derived observations might become decoupled from soil moisture deeper in the profile where it is taken up by plants (Capehart
and Carlson, 1997; Carranza et al., 2018), and they might not represent the dynamics of processes deeper in the root zone
(Bassiouni et al., 2020). The availability of a growing number of relatively accurate low-cost soil moisture sensors (Mittelbach
et al., 2011) has led to an increasing number of regional soil moisture networks, where soil moisture is measured at a large
number of sites and at several depths in the profile. Such networks, in combination with satellite-derived observations, can
provide a unique insight into the link between vegetation stress, root water uptake, and soil moisture profiles. Two of those
networks, the Twenthe and Raam networks in the Netherlands, were located in the region that suffered from the 2018 European
summer drought.

High-impact extreme events such as flash floods are often associated with sloping or upland terrain (Marchi et al., 2010).
However floods and droughts can have considerable impact in lowland areas as well, even though the main hydrological
processes can differ. For the 1976 summer drought in the Hupsel Brook catchment (Brauer et al., 2018), it was found that
soil moisture anomalies develop progressively deeper over the course of the drought, reflecting a strong link to the presence
of a relatively shallow groundwater table (Teuling et al., 2013). For the same catchment, it was found that the link between
soil moisture and groundwater table at near-saturated conditions played an equally important role in determining the onset of
saturation excess runoff and flash flood response following the August 2010 extreme precipitation (Brauer et al., 2011). In
larger lowland rivers, low topographic and hydraulic gradients can induce flooding due to backwater effects (Geertsema et al.,
2018). Due to the strong human influence on hydrological processes due to, for instance, changes in drainage density or land
use shifts towards more urbanisation, lowland areas might also be sensitive to changes in hydrological extremes (Pijl et al.,
2018).





In this study, we combine data from the Twenthe and Raam soil moisture networks located in The Netherlands with satellite derived vegetation indices (NIRv and VOD) reflecting vegetation productivity, to study regional-scale development of the 2018 agricultural drought in a lowland area during the summer months June, July and August. Specifically, we aim to i) analyse the temporal evolution of drought in the unsaturated zone in relation to the non-drought years of 2016 and 2017, ii) link dynamics of vegetation productivity to soil moisture and iii) infer the critical soil moisture content marking the transition between non-stressed and stressed soil moisture regimes and its dependency on monitoring depth.

## 2 Methods

The Raam and Twenthe soil moisture networks in The Netherlands were originally installed as validation sites for satellite-derived data products (Benninga et al., 2018; Dente et al., 2011). The Raam network faces—by Dutch standards and in comparison to the Twenthe network—substantial water shortages during normal summers (Benninga et al., 2018). This can be mainly attributed to the mostly sandy soils in the Raam network, whereas the Twenthe network is located in an area with sandy to more loamy soils. Both areas have a land cover consisting of cultivated or natural grassland, agricultural fields (maize, onion, chicory, sugar beets), and some forested sites (though these are not instrumented).

Overlapping soil moisture observations for both networks were available for 2016–2018 at discrete depths below the soil surface (5, 10, 20, 40 cm for Raam and Twenthe, and additionally 80 cm for Raam) from which daily averaged volumetric soil moisture ($\theta$ [m$^3$$_{\text{water}}$ m$^{-3}$$_{\text{soil}}$]) was obtained. The 31 days moving means of 2016 and 2017 were averaged to represent baseline conditions (referred to as climatology hereafter). The anomaly is defined as the difference between 2018 and the climatology. We assumed that measurements on 5, 10, 20, 40 and 80 cm represent the soil column between 2.5–7.5, 7.5–12.5, 12.5–27.5, 27.5–52.5 and 52.5–107.5 cm depth respectively. Stations were selected based on maximum available daily averaged data between May 2016 and September 2018 (filled circles in Figs. 1c–d). For further details on the network and sites we refer to the relevant papers (Benninga et al., 2018; Dente et al., 2011).

Photosynthetically-active radiation normalized solar-induced fluorescence (SIF, v27) was used as proxy for GPP and obtained from the GOME-2B instrument onboard the MetOp-B satellite as described in (Joiner et al., 2013, 2016) on a monthly average and $0.5 \times 0.5°$ spatial resolution. Daily NIRv was obtained by multiplication of the normalized difference vegetation index (NDVI) and total scene near-infrared reflectance (NIR$_{\text{T}}$) (Badgley et al., 2017) retrieved from the merged product from the MODIS Aqua and Terra satellites on a $0.05 \times 0.05°$ spatial resolution (Schaaf and Wang, 2015). The surface reflectance was BRDF-adjusted (Bidirectional Reflectance Distribution Function), and all values below 0 were removed. This ensured that the NIRv product has a higher spatial and temporal resolution than the SIF dataset. Although the NIRv product is relatively new, several studies highlighted the usability of this dataset. Badgley et al. (2019) showed that the relationship between NIRv and GPP was consistently linear across all values of GPP, both during drought events and during acute stress events at short timescales. Additionally, Baldocchi et al. (2020) concluded that NIRv is able to correctly represent photosynthesis across different temporal scales. Vegetation optical depth (VOD) values were obtained from Moesinger et al. (2019). VOD is used to compare with NIRv values, and to test the robustness of our analysis. VOD is a measure for above ground vegetation water





content (Konings et al., 2016; Moesinger et al., 2020), derived from space-borne microwave sensors. For our analysis, we selected the C-band to calculate the anomalies. The climatology and anomaly of NIRv and VOD were calculated similarly to $\theta$.

To infer the critical soil moisture ($\theta_{\text{critical}}$), NIRv anomaly as a function of $\theta$ was fitted by employing a piece-wise linear
function, which renders an inflection point indicating the transition between an energy-limited to a water-limited evapotranspiration regime (Seneviratne et al., 2010). We investigated the dependence of $\theta_{\text{critical}}$ on observation depth. Using bootstrapping, we determined the 5–95% uncertainty range of the inferred critical soil moisture value at each integration depth. Daily precipitation and potential ET (calculated by the KNMI with Makkink (1960)) were obtained from the KNMI stations in Volkel (06375) and Enschede (06290, see locations in Fig. 1c-d). Gridded precipitation was obtained from E-OBS (v20.0e, Cornes
et al., 2018).

## 3  Results

The strong reduction in precipitation over June and July that was centered around the Netherlands (Fig. 1a) coincided with strong negative anomalies in vegetation productivity. Coarse-scale estimates of productivity based on solar-induced fluorescence (Fig. 1b) show large negative anomalies, in particular in the eastern part of the country where soils are more sandy and
groundwater tables deeper. Higher resolution NIRv imagery shows a similar pattern, including slightly larger negative anomalies in Twenthe compared to more moderate anomalies in the Raam (Figs. 1c–d). This shows that the soil moisture networks where located at a prime location to monitor the impact of the 2018 drought.

The temporal dynamics of the vegetation productivity and soil moisture reveals considerable complexity in the response to the drought. During initial stages of the drought, NIRv kept pace with, or even sometimes exceeded, the climatological values in
both networks (Figs. 2a–b). In the beginning of summer, the NIRv anomalies are around zero for Twenthe and slightly positive for Raam, and are follwed by a sharp decline in productivity in late June. At the end of July, maximum NIRv anomalies correspond to $-30\%$ (Twenthe) and $-25\%$ (Raam). In contrast to the NIRv anomalies, soil moisture observations reveal a steady decline from the beginning of summer up to the end of July. Anomalies are found to be largest at the end of July. NIRv and soil moisture anomalies remain strongly negative in Twenthe until the beginning of October, whereas the Raam shows a
faster recovery. VOD shows a similar response as NIRv, yet the VOD anomalies exceed the climatological values during the start of the summer. The moment where the anomalies decline matches NIRv, yet the VOD anomalies recover later in the year than the NIRv anomalies.

When the dynamics of the vegetation indices during the 2018 drought are evaluated against soil moisture averaged over different depths, a strong nonlinear response becomes apparent (Fig. 3). The response is described by a piecewise linear model
with a right-hand part with zero slope (i.e. assuming no stress). This 3-parameter model describes the response better than a 2-parameter linear model as indicated by consistently higher values for the adjusted $R^2$ (average $R^2$ of 0.82 versus 0.63, see Table 1). Due to the difference in dynamics in VOD, we removed the first days of June, as the VOD anomalies were still increasing over this period (see Fig. 2). Over the selected period, VOD anomalies show no clear trend, and the average value (and period)



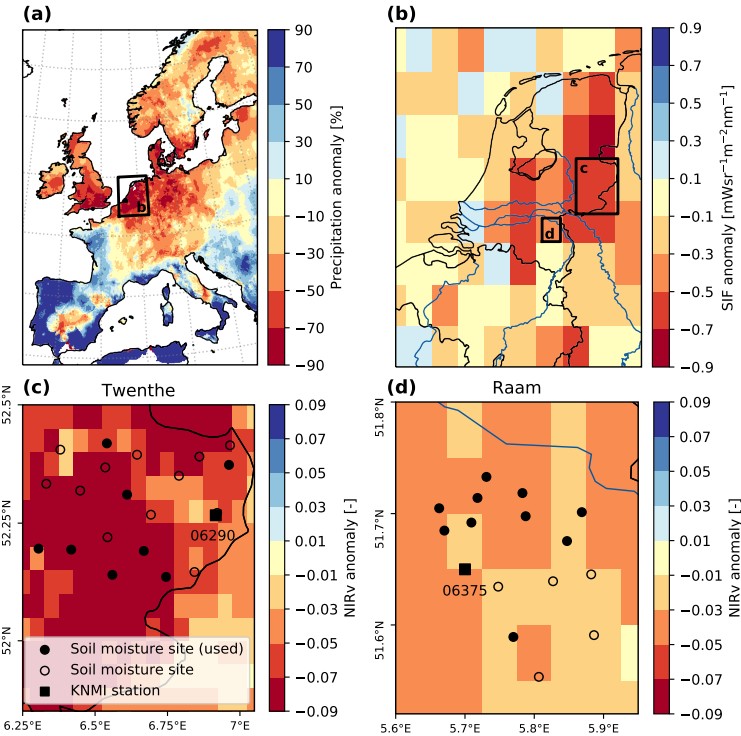

**Figure 1.** Distribution of the 2018 summer drought and vegetation productivity with respect to 2016 and 2017. Drought distribution in western Europe (a) is expressed by the relative June-July precipitation anomaly, showing that the eastern part of the Netherlands was one of the worst hit areas. This is confirmed by a similar pattern in GOME-2 SIF anomalies (b). MODIS NIRv (c,d) shows a similar distribution but at much higher spatial resolution for Twenthe (c) and Raam (d). The circles indicate in situ soil moisture measurement sites with (filled) and without data of sufficient quality (open); when filled these are included in the analyses. KNMI stations Twenthe (06290) and Volkel (06375) are indicated with black squares.

can be found in Fig. 2. Initially, NIRv and VOD anomalies remain roughly at a constant level while soil moisture decreases
considerably. This is followed by a second phase in which NIRv and VOD anomalies decrease approximately linearly with soil moisture, indicating a strong drought impact on vegetation productivity. The nonlinearity is present when soil moisture is evaluated over different depths ranging from a shallow top layer (0–5 cm) to most of the root zone (0–80 cm), using the representative soil column thickness (see Methods) to correct the soil moisture values. However, soil moisture values, including the transitional point marking the start of the drought impact on vegetation productivity, are generally lower with a difference
in volumetric water content between 0.05 and 0.10 for both sites. The point separating the two phases of non-stressed and water-stressed conditions can be interpreted as the critical soil moisture content.

Further analysis of the evolution of regional-scale average soil moisture profiles (Fig. 4) reveals the origin of the differences found in Fig. 3. In normal years, soil moisture dries out considerably in the upper layers (down to values in the range 0.15–

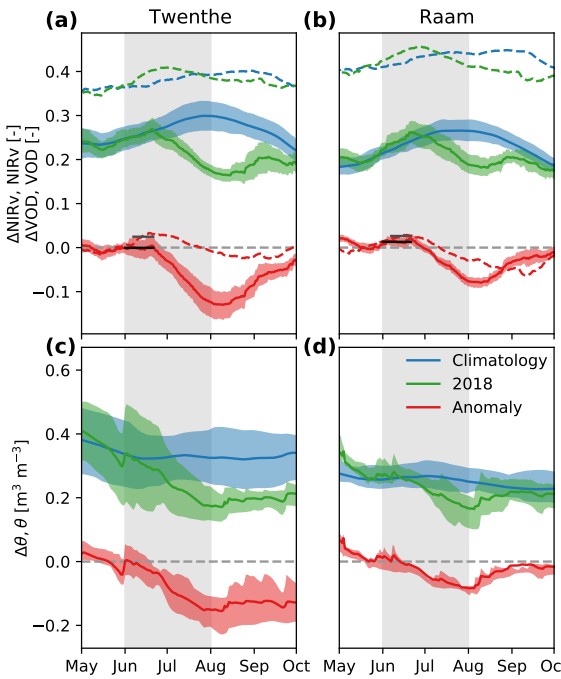

**Figure 2.** Temporal evolution of the 2018 agricultural drought. Top panels show NIRv (solid) and VOD (dashed), and bottom panels the soil moisture conditions over the growing season for Twenthe (a,c) and Raam (b,d). Soil moisture is the average observed at 40 cm depth. Horizontal lines in panels a and b indicate the non-stressed NIRv (black) and VOD (grey) values used in Fig. 3. Shaded areas indicate the 20–80% range across NIRv pixels and soil moisture stations.

0.20), but much less in the lower layers where values stay around 0.30. This is partly due to the fact that in a normal summer,
JJA potential evapotranspiration according to Makkink method (2.9–3.1 mm d$^{-1}$) is nearly balanced by precipitation with 2.3–2.8 mm d$^{-1}$. This likely allows vegetation to take up most of the water in the upper part of the root zone. In 2018, the increased atmospheric demand for evaporation as reflected in a higher potential evapotranspiration (3.6–3.7 mm d$^{-1}$, so a 20% increase), combined with a strong reduction in precipitation (1.3–1.4 mm d$^{-1}$, so a nearly 50% reduction) led to a strong initial drying of the surface layer. This is reflected in the negative anomalies which peak around the start of July (DOY 184 and 192 for Raam
and Twenthe, respectively). Only later, strong negative anomalies developed deeper in the root zone (DOY 220 and 221 for Raam and Twenthe, respectively), potentially due to enhanced root water uptake to (partly) compensate for the reduced uptake in the surface layers. This contrasts sharply with normal summer conditions where most of the uptake takes place in the surface layers. The anomalies at 40 and 80 cm depth reach their maximum only at the end of the main drought or even later. This explains the large discrepancy between surface and root zone soil moisture at the early stages of the drought.
When the critical moisture contents inferred in Fig. 3 are evaluated against the integration depth of the soil moisture observations, we find the results in Fig. 5. Ideally, there should be no dependency of the critical moisture content on depth, because this would facilitate the identification and use of the critical moisture content in models. However both networks show a similar





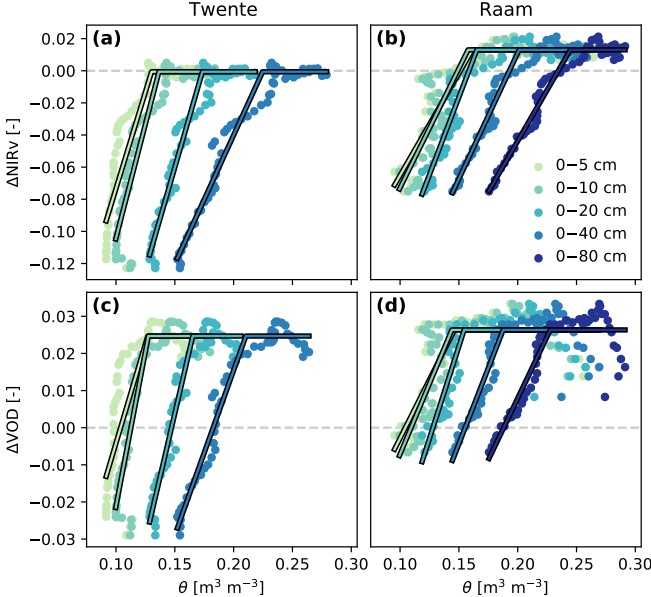

**Figure 3.** Relation between anomalies in vegetation productivity and soil moisture (dots) and the piecewise linear fit (lines) for Twenthe (a, c) and Raam (b, d), for both NIRv (a, b) and VOD (c, d). The horizontal part of the piecewise fit was set at the average vegetation index anomaly value in the first part of the summer period (corresponding to the horizontal line in Fig. 2a–b).

strong dependency with depth, with the inferred critical moisture content ranging from 0.13–0.16 $m^3_{water}$ $m^{-3}_{soil}$ for shallow soil moisture observations, to over 0.20 $m^3_{water}$ $m^{-3}_{soil}$ when observations over most of the root zone are used. The inferred relations between critical soil moisture and depth are found to be roughly equal for the fits based on NIRv and VOD data. The uncertainty bars resulting from bootstrapping show larger uncertainty at shallower integration depths, yet the values found at shallower depths are lower than values at deeper integration depths. The critical soil moisture values can be found in Table 1. This table also shows the relative $\theta_{critical}$ determined using the minimum and maximum soil moisture values over the period 2016–2018. Because the root zone is presumably deeper than 1 m, it is possible that observations over the entire root zone will lead to even higher values.

## 4   Discussion

This study combined data from two Dutch soil moisture networks with high-resolution satellite vegetation indices as a novel approach to quantify agricultural drought conditions and impact. The 2018 summer drought had considerable impact in the areas where the networks were situated.

The inferred $\theta_{critical}$—marking the transition between non-stressed (energy limited ET) and stressed (water-limited ET) soil moisture regimes—is found to be dependent on monitoring depth. Accurate determination of the $\theta_{critical}$ is essential for describing the relation between vegetation's response to water stress and carbon flux predictions during drought events (Boese





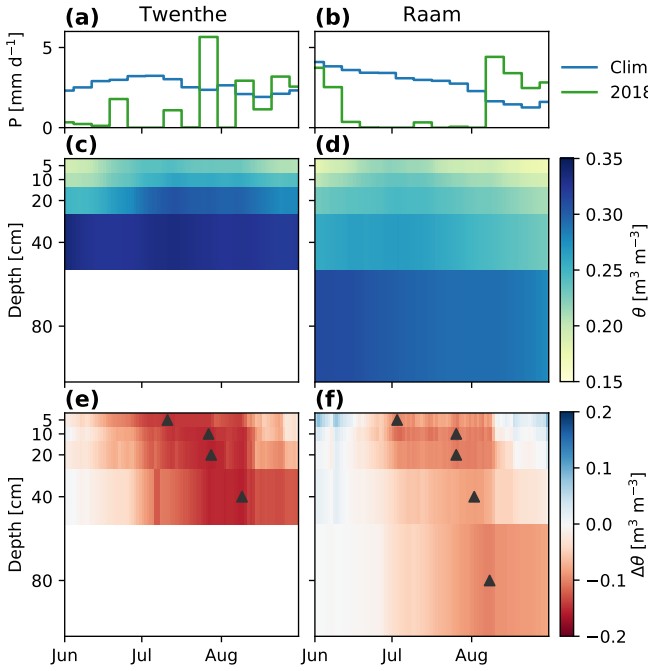

**Figure 4.** Temporal evolution of observed precipitation and soil moisture profiles during the 2018 drought. For precipitation, the top panels (a and b) show the precipitation recorded at the KNMI stations of Twenthe and Volkel (see location in Fig. 1c/d). For soil moisture, the climatology (mean 2016–2017, panels c and d) and the 2018 anomalies (panels e and f) are shown for Twenthe (left panels) and Raam (right panels). The triangles in panels e and f indicate the moment of maximum negative anomaly at each depth.

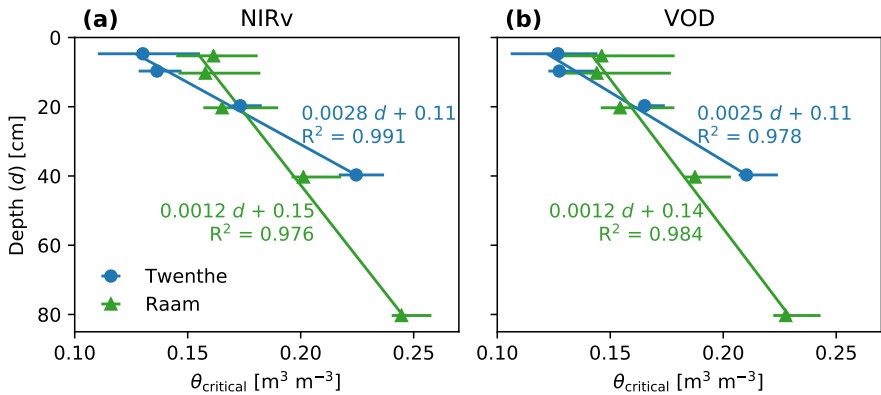

**Figure 5.** Relation between critical soil moisture and the integration depth (denoted as $d$ in the equation) of soil moisture used in the inference. Panel a shows the relation based on the NIRv data, and panel b shows the relation based on the VOD data. Horizontal lines indicate the 5–95% range of critical soil moisture values.





**Table 1.** Fit statistics and resulting critical soil moisture content based on both NIRv and VOD data. $R^2_{adjusted}$ values are shown for both the piecewise (pw) and linear (lin) fits, adjusted for the number of parameters used in the fit, the value between brackets shows the standard $R^2$ value. The critical soil moisture content between brackets is the value normalized between minimum and maximum moisture content values at each integration depth.

|  | Depth [cm] | NIRv | | | VOD | | |
|---|---|---|---|---|---|---|---|
|  |  | $R^2$pw | $R^2$lin | $\theta_{critical}$ | $R^2$pw | $R^2$lin | $\theta_{critical}$ |
| Raam | 5 | 0.80 (0.81) | 0.61 (0.62) | 0.16 (0.24) | 0.55 (0.57) | 0.24 (0.27) | 0.15 (0.18) |
|  | 10 | 0.82 (0.83) | 0.65 (0.66) | 0.16 (0.22) | 0.58 (0.61) | 0.28 (0.30) | 0.14 (0.17) |
|  | 20 | 0.91 (0.91) | 0.72 (0.73) | 0.17 (0.20) | 0.73 (0.74) | 0.34 (0.36) | 0.15 (0.16) |
|  | 40 | 0.96 (0.96) | 0.82 (0.82) | 0.20 (0.29) | 0.83 (0.84) | 0.46 (0.48) | 0.19 (0.22) |
|  | 80 | 0.97 (0.98) | 0.89 (0.89) | 0.24 (0.37) | 0.85 (0.86) | 0.57 (0.58) | 0.23 (0.29) |
| Twenthe | 5 | 0.72 (0.73) | 0.59 (0.60) | 0.13 (0.08) | 0.60 (0.63) | 0.48 (0.50) | 0.13 (0.08) |
|  | 10 | 0.86 (0.87) | 0.69 (0.70) | 0.14 (0.08) | 0.82 (0.83) | 0.62 (0.63) | 0.13 (0.06) |
|  | 20 | 0.95 (0.96) | 0.82 (0.83) | 0.17 (0.11) | 0.95 (0.95) | 0.79 (0.80) | 0.17 (0.09) |
|  | 40 | 0.97 (0.97) | 0.88 (0.88) | 0.22 (0.18) | 0.97 (0.97) | 0.84 (0.84) | 0.21 (0.15) |

et al., 2019; Green et al., 2019; Stocker et al., 2019) as current parametric expressions are unsuitable under droughts (Madi et al., 2018). This study highlights the particular value of in situ soil moisture networks, besides their purpose to calibrate and validate satellite-derived observations (Dorigo et al., 2011), to inform about $\theta_{critical}$, as root water uptake dynamics and ET rates cannot easily be derived from satellite observations (Purdy et al., 2018).

We found a decline in NIRv and VOD to occur only once surface soil moisture had already reached its lowest level. Satellite-derived observations of the soil's subsurface can certainly serve as early predictors for drought onset (Ford et al., 2015; Otkin et al., 2018), yet drought also leads to decoupling of the soil moisture signal over depth (Carranza et al., 2018), rendering satellite-derived soil moisture or in situ surface soil moisture observations uninformative about root water uptake and drought impact status. This effect, in combination with the sandy texture of the soils in both networks, can also explain why we find values for $\theta_{critical}$ that are lower than those from recent estimates based on satellite soil moisture (Denissen et al., 2020). Assessment of vegetation response to profile soil moisture requires observations both at multiple depths and at multiple profiles to average out small-scale heterogeneities (Teuling et al., 2006).

This study determined the $\theta_{critical}$ with already available data. The method would in principle allow to identify root water uptake regimes during droughts without the need for (difficult to obtain) vegetation-driven biophysical landscape interactions (for example (Prentice et al., 2014; Warren et al., 2015; van der Ploeg et al., 2018)). However, within the Raam and Twenthe networks the maximum measurement depth of $\theta$ (80 and 40 cm respectively) may have been insufficient to capture the complete propagation of soil moisture anomalies in the root zone, and their possible link to root water uptake dynamics. Were the measurement set ups of both networks harmonized by covering the entire rootzone, it would have provided a more accurate comparison of drought impacts and variability in soil moisture (Dorigo et al., 2011). When the focus of establishing a soil





moisture network is not to validate satellite-derived observations—as was the case for these two networks—, but quantifying drought effects on root water uptake, the maximum rooting depth of the vegetation near soil moisture stations should be considered, even though temporal dynamics of soil moisture and root water uptake under non-drought conditions predominantly occur in the upper 70 cm of the soil profile (Teuling et al., 2006).

Ideally, the values for $\theta_{\mathrm{critical}}$ are considered with respect to the wilting point and field capacity, because these, in concert with the rooting depth, determine the soil moisture dynamics (Albertson and Kiely, 2001). However these values themselves are highly variable spatially but also vertically over the soil profile. For sandy (Raam) to more loamy (Twenthe) soils, these characteristic soil moisture values are generally assumed to be in the range of a few vol. % (wilting point) and between 15 and 25 vol. % (field capacity). However differences between various pedotransfer methods can be large (Teuling et al., 2009). Based on the length of the time before a reduction in NIRv and VOD anomalies was first observed, it can be inferred that even in these coarse soils, a significant storage exists between field capacity and the critical moisture content that can be utilized by plants during drought onset.

This study also provides realistic environmental conditions of drought at relevant scales. In a recent meta-analysis of studies on drought impacts on ecosystems, Slette et al. (2019) concluded that drought is often poorly defined, and many supposed drought experiments take place within the normal range of climate variability rather than an extreme drought. This is problematic because drought impact is not proportional to drought severity, but increases rapidly once a critical threshold has been exceeded. More research is therefore needed to identify and quantify drought thresholds and impacts across ecosystems and climate regions, especially in light of co-evolution in soil-vegetation-fauna-microbial relations, particularly the different strategies in which these relationships are adopted, modified or adapted (Robinson et al., 2019). Failure to represent such ecosystem strategies in Earth system models might affect our ability to make reliable projections of future drought impact. The methodology presented here informs to better constrain drought-relevant parameters, such as the critical moisture content, in models.

## 5 Conclusions

A prolonged period of no (or very low) precipitation during the summer of 2018 caused profound negative soil moisture anomalies compared to the two prior years in the Raam and Twenthe. The decrease in soil moisture proceeded into deeper layers with time as a consequence of root water uptake shifting predominantly to those layers. Subsequently, ET decreased, which is in line with the low 2018 GPP proxies SIF, NIRv and VOD obtained via satellites throughout the growing season. Root water uptake was observed to shift to deeper layers after the first reduction in NIRv and VOD, indicating that changing root water uptake patterns can help to reduce drought impact, but not to avoid it in the case of the drought of 2018. Soil moisture, ET, and GPP remained low until the end of summer.

Using a novel approach, the critical soil moisture content ($\theta_{\mathrm{critical}}$) was derived from NIRv and VOD anomalies and soil moisture measurements on multiple depths. This nonlinear relation reflects the observation that negative soil moisture anomalies develop weeks before the first reduction in vegetation indices. The critical soil moisture content in the Raam on 40 cm





depth is found to be 0.19 and in Twenthe 0.22 [$m^3_{water}$ $m^{-3}_{soil}$]. The apparent critical soil moisture content increased with depth and this relationship was shown to be linear. The critical soil moisture content can serve as indicator to mark the transition between non-stressed and stressed conditions to examine the impact on the gross primary productivity of vegetation and effect on the carbon cycle in models during droughts.

*Data availability.* Daily precipitation, potential evaporation, NIRv and average soil moisture data for the different depths over the period
2016–2018 for Raam and Twenthe can be obtained at https://doi.org/10.6084/m9.figshare.12090591. E-OBS gridded precipitation (v20.0e) was obtained from www.ecad.eu. The MODIS NDVI and $NIR_T$ was obtained from https://lpdaac.usgs.gov/.

*Author contributions.* AMS carried out the original study under supervision of AJT, MvdP, and NES. AJT conceived and coordinated the study. CDUC, FvdB, HJFB, and RvdV assisted with the collection and interpretation of the soil moisture data. GK helped with the processing and interpretation of the satellite data. JB verified and extended the analysis it with VOD data and produced the final figures and results. AJT
and MvdP drafted the manuscript; JB critically revised the manuscript; All authors gave final approval for publication and agree to be held accountable for the work performed therein.

*Competing interests.* The authors declare that they have no competing interests.

*Acknowledgements.* We acknowledge the E-OBS dataset from the EU-FP6 project UERRA (http://www.uerra.eu) and the Copernicus Climate Change Service, and the data providers in the ECA&D project (https://www.ecad.eu)





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
