# Peer review of "Anatomy of the 2018 agricultural drought in The Netherlands using in situ soil moisture and satellite vegetation indices"

_Hydrology and Earth System Sciences, 2020_

## Referee Comment (RC1) · Anonymous Referee #1 · 14 Sep 2020

The authors present a very interesting and for the scientific community (hydrology and remote sensing, and agricultural science, modelling and resources management in a broader sense) highly relevant study on the spatiotemporal assessment of the critical soil moisture content in different soil depths and based on selected space-borne vegetation indices, and compared with in situ data from two soil moisture measuring networks (Raam and Twente) in The Netherlands. The focus is on the drought and heat event in the spring and summer months of 2018, which was particularly evident in northern and central parts of Europe with below-average rainfall and above-average temperatures. The derivation of the thematic superstructure, the description of materials and methods as well as the presentation of results and subsequent discussion

follows a clear logical structure. The writing/language is precise and stylistically confident. Sometimes some sentences are too long, which makes them difficult to read. The conclusions are clearly understandable and unambiguous. All in all, I would like to state that I enjoyed reading the manuscript very much. Except for a few revisions that I think are necessary (see below points 9 to 14), the manuscript is in a condition worthy of publication. I would like to thank you for the opportunity to review the manuscript and would like to recommend publication of the article in HESS.

1. Does the paper address relevant scientific questions within the scope of HESS? The authors present a very interesting and for the scientific community (hydrology and remote sensing in the narrower sense) highly relevant study on the spatiotemporal assessment of the critical soil moisture content in different soil depths and based on selected remote sensing based vegetation indices for two soil moisture measuring networks (Raam and Twente) in The Netherlands. The focus is on the drought and heat event in the spring and summer months of 2018, which was particularly evident in northern and central parts of Europe with below-average rainfall and above-average temperatures. In my own opinion, the study is thus fully within the scope of the HESS Journal and is therefore likely to be of great interest to readers from the fields of agricultural science and modelling as well as, in a broader sense, resources management (e.g., scope and need for action in context of one central finding by authors that "[. . .] negative soil moisture anomalies develop weeks before the first reduction in vegetation indices.", lines 243-244).

2. Does the paper present novel concepts, ideas, tools, or data? Indeed, the authors present a new approach to the determination of the critical soil moisture content by means of highly temporally resolved daily remote sensing data. Not only the approach as such is new, but also the input data (NIRv and VOD) are up-to-date.

3. Are substantial conclusions reached? The provided conclusions are precisely presented and clearly understandable. The objectives mentioned in the introductory chapter are answered in sufficient detail.

4. Are the scientific methods and assumptions valid and clearly outlined? To my best knowledge, the scientific methods used and assumptions are adequately chosen and clearly outlined.

5. Are the results sufficient to support the interpretations and conclusions? Yes, absolutely. As already mentioned below, the entire manuscript follows a logical and clear structure. This also applies to the presentation of the results (textual as well as in the form of the five illustrations and a table). The interpretation of the results and the conclusion of the findings is coherent and comprehensible. No generalizing statements are made without reference to the study.

6. Is the description of experiments and calculations sufficiently complete and precise to allow their reproduction by fellow scientists (traceability of results)? The description of the experimental/study setup is given in full. The information on the data sources utilized is complete.

7. Do the authors give proper credit to related work and clearly indicate their own new/original contribution? The findings from other studies are contextually related and clearly recognizable. A corresponding differentiation to new/original contributions is possible.

8. Does the title clearly reflect the contents of the paper? In my understanding, the title of this study adequately summarizes the content. The title is clear and interestingly designed. In my opinion, a change of the title is not necessary.

9. Does the abstract provide a concise and complete summary? The abstract is a concise, precise and more or less complete summary of the work. Only the definitions of the abbreviations (see note under point 12) are missing. For a complete summary in my opinion, information on the most important data sources (e.g. MODIS product, spatiotemporal resolution) and most important results should be given in figures (e.g. fit statistics and critical soil moisture).

[Figure]

10. Is the overall presentation well structured and clear? The manuscript follows a clear logical, causal structure, also in accordance with HESS guidelines. The central theme of the study is recognizable throughout the manuscript, from the introduction to the topic to the conclusion. Unnecessary repetitions are not visible in the text. Nevertheless, I recommend a slight shortening of the introductory chapter up to a maximum of 1.5 to 2 pages.

In my understanding, the sentence "The availability of a [...]" (line 71) marks a new paragraph.

11. Is the language fluent and precise? The authors present a very concise and well written, and logically structured manuscript. The language is fluent and precise without major exceptions.

Only some sentences seem to be too long, so that the readability is a bit difficult. For this reason, I recommend a revision regarding the shortening of some sentences or separation of one sentence into two (e.g., lines 17-19, 39-41).

In line 48, the sentence "This is confirmed [...]" is missing the preposition "by".

In line 85, the reading flow of the sentence is a bit hampered by double "due to". I recommend restructuring this sentence.

12. Are mathematical formulae, symbols, abbreviations, and units correctly defined and used? Mathematical functions are not included in the manuscript. Although the mathematical function for deriving the daily NIRv index would certainly be beneficial for the reader, this is not urgently necessary due to the comprehensive reference.

The listed symbols are introduced accordingly and used consistently in the text. An indexing of the unit of volumetric water content (water/soil) is not necessary in my opinion, but it is also not negatively noticeable.

The listed abbreviations are introduced adequately and are used consistently in the text. Only for the abstract, a definition of the abbreviations NIRv, VOD and GPP according to the HESS guidelines has to be implemented.

13. Should any parts of the paper (text, formulae, figures, tables) be clarified, reduced, combined, or eliminated? As mentioned above, the structure of the manuscript follows a clearly recognizable red thread. For the chapter 'Introduction', a slight reduction of the text to a maximum of 1.5 to 2 pages is recommended. On the other hand, some additions in the form of examples would certainly be advantageous. A list of two or three examples of mentioned 'remote sensing products' (line 39) and corresponding references should be added in my opinion. Also specifications on the temporal resolutions for NIRv, VOD, and SIF data should be added – for instance using brackets – even though it is obvious from the section 'Material and Methods'. Furthermore, the basis for the assumption in lines 105 to 106 is not clearly evident. What is the basis of this assumption? How are the soils in Raam and Twente soil moisture networks characterized?

For the inexperienced reader, information on the area size of the individual Raam and Twente networks and the density of the networks would be helpful. How far apart are the individual stations located in each area? Are the soil moisture networks heterogeneous in terms of topography? In my opinion, an appropriate characterization of the areas would increase the readability of the results.

Moreover, a specification of the space-borne microwave sensor (line 121; AMSR-E and AMSR2, WindSat?)

14. Are the number and quality of references appropriate? In total, 64 references providing actual information from a diversity of international high-ranking journals is listed. The references are appropriate and adequately support the content of manuscript. The list of references is complete, meaning that all citations in the text are provided in the list too. The style of citations and references is in agreement with the guidelines of HESS. Only the abbreviations of the journals according to the guidelines (Journal Title Abbreviations by Caltech Library) should still be implemented by the authors.

[Figure]

15. Is the amount and quality of supplementary material appropriate? There is no supplementary material provided.

Please also note the supplement to this comment:
https://hess.copernicus.org/preprints/hess-2020-358/hess-2020-358-RC1-supplement.pdf

────────────────────────

---

## Referee Comment (RC2) · Anonymous Referee #2 · 6 Oct 2020

This is a nice work that show the relation found between study in situ soil moisture profile (SMC) measurements of the Raam and Twenthe networks in the Netherlands, with two satellite derived (RS) vegetation indices (VIs), NIRv and VOD, during the 2018 summer.

I believe that this manuscript has the quality standards of the journal and presents a very interesting work combining field measures with remote sensing measures. This is an important point. However, I have a few comments to the authors so the reader will find it easier to understand:

1) In the abstract you used a lot of acronyms and that is ok. But when you say "and

its effect on GPP in models" I suggest to put first what is that GPP. You use NIRv and VOD but you say that they are vegetation indexes and that is fine, but what is GPP?

2) Deeper are the measures in soil are this really reflected by the VIs? or this is just a consequence of the correlation among depths of SMC. For me, it is hard to see that a measure at 80 cm depth can be reflected in RS bands. But the measures between 80 cm and 10 cm can be correlated. Can you show this correlations among depths?

3) Are precipitation anomaly and SIF anomaly calculated in the same way that NIRv anomaly?

4) You really have three years. Calculating these anomalies means that you have the average of two years and then compare it with 2018. Is it right? Perhaps you should describe 2016 and 2017 as quite normal years, otherwise it looks too few years to consider the estimation a week anomaly.

5) Figure 2. This figure is very important to understand this nice work. You should improve it as you talk about black lines (almost I cannot see it), dashed lines, etc. Please, make it more clear. I imaging that this is the average of an area. Isn't it? If I understood it right just indicate it in the label of this figure. You mention in this label Figure 3. I think that you shouldn't. Another point is that if you improve Figure 2 then the data use from Figure 2 in Figure 3 will be easier to understand.

6) Figure 3. You mention in the label "vegetation productivity". What I can see is the relation between VIs anomalies with SWC. Between VIs and vegetation productivity which is the relation? This relation is using a time lag of 0. Did you try the relation with some time lag of 1 or 2? You mention in the introduction the lag that exist between meteorological anomalies and VIs anomalies. Exist any lag between SWC and VIs?

7) In table 1 you show the normalized critical soil moisture content in brackets. I believe that will be more interesting to see the s.e. of this estimation.

8) In the abstract you said the nonlinear relation between negative soil moisture anomalies and VIs reflects that the drought was develop weeks before the first reduction in vegetation indices. Perhaps you should explore how many weeks before.

Finally, this "anatomy of" expression in the title I will change it for other or just suppressed it.

I really enjoyed your work.

———————————————

---

## Author Comment (AC1) · 14 Oct 2020

We would like to thank Anonymous Referee #1 (AR1) for their positive and concise comment, and for the recommendation for publication in HESS. Below, we will respond to the comments made by AR1 which require explanation or additional information: the comments from AR1 in black, our response in blue.

9. Only the definitions of the abbreviations (see note under point 12) are missing. For a complete summary in my opinion, information on the most important data sources (e.g. MODIS product, spatiotemporal resolution) and most important results should be given in figures (e.g. fit statistics and critical soil moisture).
We agree that we did not sufficiently explained every abbreviation, and we will update this in the next version of the manuscript.

10. Nevertheless, I recommend a slight shortening of the introductory chapter up to a maximum of 1.5 to 2 pages.
We understand the concerns of AR1 that the current introduction is too lengthy. We will try to shorten some paragraphs, but we believe that the current introduction covers all relevant topics necessary to introduce and frame our work.

In my understanding, the sentence "The availability of a [. . .]" (line 71) marks a new paragraph.
This is indeed correct, and we will correct this.

11. Only some sentences seem to be too long, so that the readability is a bit difficult. For this reason, I recommend a revision regarding the shortening of some sentences or separation of one sentence into two (e.g., lines 17-19, 39-41).
We will shorten these sentences together with the shortening of the introduction.

In line 48, the sentence "This is confirmed [. . .]" is missing the preposition "by".
Thanks for this correction, we will fix this sentence.

In line 85, the reading flow of the sentence is a bit hampered by double "due to". I recommend restructuring this sentence
We will rewrite this sentence.

12. Although the mathematical function for deriving the daily NIRv index would certainly be beneficial for the reader, this is not urgently necessary due to the comprehensive reference.
The calculation procedure to calculate NIRv is described in lines 111-112. We will add this as equation to the text.

Only for the abstract, a definition of the abbreviations NIRv, VOD and GPP according to the HESS guidelines has to be implemented.
We will make sure every abbreviation is explained before its introduction.

13. For the chapter 'Introduction', a slight reduction of the text to a maximum of 1.5 to 2 pages is recommended. On the other hand, some additions in the form of examples would certainly be advantageous. A list of two or three examples of mentioned 'remote sensing products' (line 39) and corresponding references should be added in my opinion. Also specifications on the temporal resolutions for NIRv, VOD, and SIF data should be added – for instance using brackets – even though it is obvious from the section 'Material

and Methods'. Furthermore, the basis for the assumption in lines 105 to 106 is not clearly evident. What is the basis of this assumption? How are the soils in Raam and Twente soil moisture networks characterized?

As mentioned above, we will try to shorten the introduction. The "remote sensing products" in line 39 refers to the products used in the studies mentioned before this sentence. The representative soil depth assumption is based on that the measure value is representative for the soil around this depth: the probe at 5 cm depth is representative for a total depth of 5 cm: from 2.5cm to 7.5cm, et cetera.

14. For the inexperienced reader, information on the area size of the individual Raam and Twente networks and the density of the networks would be helpful. How far apart are the individual stations located in each area? Are the soil moisture networks heterogeneous in terms of topography? In my opinion, an appropriate characterization of the areas would increase the readability of the results.

The location of the soil moisture sensors, and the rough area of each region can be found in Figure 2. To give a sense of the coverage of each network, we will add information about the mean spacing for both Twenthe (6.2 km) and Raam (3.4 km). We will add some information on the topography, further details on the networks are described in the first paragraph of the methods.

Moreover, a specification of the space-borne microwave sensor (line 121; AMSR-E and AMSR2, WindSat?)

The data is based on multiple sensors: SSM/I, TMI, AMSR-E, WindSat, and AMSR2. We will add this information.

---

## Author Comment (AC2) · 14 Oct 2020

We would like to thank Anonymous Referee #2 (AR2) for their constructive and positive comments. Below, we will respond to the comments made by AR2: the comments from AR2 in black, our response in blue.

This is a nice work that show the relation found between study in situ soil moisture profile (SMC) measurements of the Raam and Twenthe networks in the Netherlands, with two satellite derived (RS) vegetation indices (VIs), NIRv and VOD, during the 2018 summer. I believe that this manuscript has the quality standards of the journal and presents a very interesting work combining field measures with remote sensing measures. This is an important point. However, I have a few comments to the authors so the reader will find it easier to understand:

Thanks for writing the review and the positive feedback. Below we will respond to the comments to explain and/or elaborate.

1) In the abstract you used a lot of acronyms and that is ok. But when you say "and its effect on GPP in models" I suggest to put first what is that GPP. You use NIRv and VOD but you say that they are vegetation indexes and that is fine, but what is GPP?

We agree with AR2, and we will make sure we explain each abbreviation at its introduction.

2) Deeper are the measures in soil are this really reflected by the VIs? or this is just a consequence of the correlation among depths of SMC. For me, it is hard to see that a measure at 80 cm depth can be reflected in RS bands. But the measures between 80 cm and 10 cm can be correlated. Can you show this correlations among depths?

It is true that the soil moisture measurements are correlated. However, we do not use RS observations of soil moisture, but use remotely sensed vegetation indices. Here, we assume that the vegetation indices reflect the state of the soil column reachable by roots, i.e. averaged over the whole root zone. Since the depth of the root zone over which the water uptake takes places is not known a priori and can even change over the course of a drought, we evaluate the relation with vegetation indices over different averaging depths. Hence when the vegetation indices show a decline, this matches with the available water in the soil column.

3) Are precipitation anomaly and SIF anomaly calculated in the same way that NIRv anomaly?

Yes those are all calculated in the same way. We will clarify this is the next version of the manuscript.

4) You really have three years. Calculating these anomalies means that you have the average of two years and then compare it with 2018. Is it right? Perhaps you should describe 2016 and 2017 as quite normal years, otherwise it looks too few years to consider the estimation a week anomaly.

This is correct, we will put some more attention on how 2016 and 2017 represent normal condition.

5) Figure 2. This figure is very important to understand this nice work. You should improve it as you talk about black lines (almost I cannot see it), dashed lines, etc. Please, make it more clear. I imaging that this is the average of an area. Isn't it? If I understood it right just indicate it in the label of this figure. You mention in this label

Figure 3. I think that you shouldn't. Another point is that if you improve Figure 2 then the data use from Figure 2 in Figure 3 will be easier to understand.

We will clarify the figures and the captions by better explaining the components of each figure, and also clarify the relation between the individual figures.

6) Figure 3. You mention in the label "vegetation productivity". What I can see is the relation between VIs anomalies with SWC. Between VIs and vegetation productivity which is the relation? This relation is using a time lag of 0. Did you try the relation with some time lag of 1 or 2? You mention in the introduction the lag that exist between meteorological anomalies and VIs anomalies. Exist any lag between SWC and VIs?

We understand the confusion cause by "vegetation productivity", as this was an effort to describe both NIRv and VOD. We will clarify this in the next version. We did not use any time lag, as this is just a scatter plot between two variables.

7) In table 1 you show the normalized critical soil moisture content in brackets. I believe that will be more interesting to see the s.e. of this estimation.

An indication of the reliability of the determined crical soil moisture is already presented in Figure 5 using the horizontal lines, which are indicative the of the s.e.. We believe it is more interesting to see how these determined values compare to the measured soil moisture values.

8) In the abstract you said the nonlinear relation between negative soil moisture anomalies and VIs reflects that the drought was develop weeks before the first reduction in vegetation indices. Perhaps you should explore how many weeks before.

An estimate between the offset between the first reduction in soil moisture and the first reduction in vegetation indices is represented by the horizontal lines in Figure 2. We will quantify this period.

Finally, this "anatomy of" expression in the title I will change it for other or just suppressed it.

We believe using this wording nicely explains how we try to understand the different dynamics of the summer drought in this region, hence the "anatomy of". Note that referee #1 noted no issues with the title, and that we have used similar titles in 2 previous HESS publications (see Brauer et al. 2011, Anatomy of extraordinary rainfall … and Geertsema et al. 2018, Anatomy of Simultaneous Flood Peaks…).

I really enjoyed your work.

Thanks for your kind words, and thanks for your time and effort in writing this constructive comment!

---

## Author Response (AR1)

**Point-to-point response**

Here we reply to each point made by the two reviewers.

**AR1**

We would like to thank Anonymous Referee #1 (AR1) for their positive and concise comment, and for the recommendation
for publication in HESS. Below, we will respond to the comments made by AR1 which require explanation or additional information: the comments from AR1 in black, our response in blue.

    1. Does the paper address relevant scientific questions within the scope of HESS? The authors present a very interesting and for the scientific community (hydrology and remote sensing in the narrower sense) highly relevant study on the spatiotemporal
assessment of the critical soil moisture content in different soil depths and based on selected remote sensing based vegetation indices for two soil moisture measuring networks (Raam and Twente) in The Netherlands. The focus is on the drought and heat event in the spring and summer months of 2018, which was particularly evident in northern and central parts of Europe with below-average rainfall and above-average temperatures. In my own opinion, the study is thus fully within the scope of the HESS Journal and is therefore likely to be of great interest to readers from the fields of agricultural science and modelling as
well as, in a broader sense, resources management (e.g., scope and need for action in context of one central finding by authors that "[. . .] negative soil moisture anomalies develop weeks before the first reduction in vegetation indices.", lines 243-244).
    Thanks for the positive evaluation.
    2. Does the paper present novel concepts, ideas, tools, or data? Indeed, the authors present a new approach to the determination of the critical soil moisture content by means of highly temporally resolved daily remote sensing data. Not only the
approach as such is new, but also the input data (NIRv and VOD) are up-to-date.
    No reply necessary.
    3. Are substantial conclusions reached? The provided conclusions are precisely presented and clearly understandable. The objectives mentioned in the introductory chapter are answered in sufficient detail.
    No reply necessary.
4. Are the scientific methods and assumptions valid and clearly outlined? To my best knowledge, the scientific methods used and assumptions are adequately chosen and clearly outlined.
    No reply necessary.
    5. Are the results sufficient to support the interpretations and conclusions? Yes, absolutely. As already mentioned below, the entire manuscript follows a logical and clear structure. This also applies to the presentation of the results (textual as well as in
the form of the five illustrations and a table). The interpretation of the results and the conclusion of the findings is coherent and comprehensible. No generalizing statements are made without reference to the study.
    No reply necessary.
    6. Is the description of experiments and calculations sufficiently complete and precise to allow their reproduction by fellow scientists (traceability of results)? The description of the experimental/study setup is given in full. The information on the data
sources utilized is complete.
    No reply necessary.
    7. Do the authors give proper credit to related work and clearly indicate their own new/original contribution? The findings from other studies are contextually related and clearly recognizable. A corresponding differentiation to new/original contributions is possible.
No reply necessary.
    8. Does the title clearly reflect the contents of the paper? In my understanding, the title of this study adequately summarizes the content. The title is clear and interestingly designed. In my opinion, a change of the title is not necessary.
    While AR2 does not fully agree with the current title, we decided to follow this advice and keep the original title.
    9. Does the abstract provide a concise and complete summary? The abstract is a concise, precise and more or less complete
summary of the work. Only the definitions of the abbreviations (see note under point 12) are missing. For a complete summary in my opinion, information on the most important data sources (e.g. MODIS product, spatiotemporal resolution) and most important results should be given in figures (e.g. fit statistics and critical soil moisture).

We agree that we did not sufficiently explained every abbreviation, and we have explained the GPP abbrevation in the abstract. Furthermore, we have ensured that the spatial and temporal resolutions are given in the Methods section for each dataset.

10. Nevertheless, I recommend a slight shortening of the introductory chapter up to a maximum of 1.5 to 2 pages.

We understand the concerns of AR1 that the current introduction is too lengthy. We have shortened the introduction and reduced complex sentences in length. However, we believe that the current introduction covers all relevant topics necessary to introduce and frame our work, which make complete removal of certain paragraphs difficult.

In my understanding, the sentence "The availability of a [. . .]" (line 71) marks a new paragraph.

This is indeed correct, and we have corrected this.

11. Only some sentences seem to be too long, so that the readability is a bit difficult. For this reason, I recommend a revision regarding the shortening of some sentences or separation of one sentence into two (e.g., lines 17-19, 39-41).

We have shortened these sentences together with the shortening of the introduction.

In line 48, the sentence "This is confirmed [. . .]" is missing the preposition "by".

Thanks for this correction, we have fixed this.

In line 85, the reading flow of the sentence is a bit hampered by double "due to". I recommend restructuring this sentence

We have rewritten this sentence.

12. Although the mathematical function for deriving the daily NIRv index would certainly be beneficial for the reader, this is not urgently necessary due to the comprehensive reference.

To improve clarity, we have included the equation used to calcualte NIRv.

Only for the abstract, a definition of the abbreviations NIRv, VOD and GPP according to the HESS guidelines has to be implemented.

We have excluded usage of the GPP abbreviation in the abstract. As for NIRv and VOD, we believe that the exact definition of each abbreviation is redundant for the abstract.

13. For the chapter 'Introduction', a slight reduction of the text to a maximum of 1.5 to 2 pages is recommended. On the other hand, some additions in the form of examples would certainly be advantageous. A list of two or three examples of mentioned 'remote sensing products' (line 39) and corresponding references should be added in my opinion. Also specifications on the temporal resolutions for NIRv, VOD, and SIF data should be added – for instance using brackets – even though it is obvious from the section 'Material and Methods'. Furthermore, the basis for the assumption in lines 105 to 106 is not clearly evident. What is the basis of this assumption? How are the soils in Raam and Twente soil moisture networks characterized?

As mentioned earlier, we have shortened the introduction. The "remote sensing products" in line 39 refers to the products used in the study mentioned before this sentence, which is now clarified. The representative soil depth assumption is based on that the measure value is representative for the soil around this depth: the probe at 5 cm depth is representative for a total depth of 5 cm: from 2.5cm to 7.5cm, et cetera.

14. For the inexperienced reader, information on the area size of the individual Raam and Twente networks and the density of the networks would be helpful. How far apart are the individual stations located in each area? Are the soil moisture networks heterogeneous in terms of topography? In my opinion, an appropriate characterization of the areas would increase the readability of the results.

The location of the soil moisture sensors, and the rough area of each region can be found in Figure 2. To give a sense of the coverage of each network, we have added information about both the mean spacing for both Twente (6.2 km) and Raam (3.4 km), and the mean elevation of the sites.

Moreover, a specification of the space-borne microwave sensor (line 121; AMSR-E and AMSR2, WindSat?)

The data is based on multiple sensors: SSM/I, TMI, AMSR-E, WindSat, and AMSR2. We have added this information.

**AR2**

We would like to thank Anonymous Referee #2 (AR2) for their constructive and positive comments. Below, we will respond to the comments made by AR2: the comments from AR2 in black, our response in blue.

This is a nice work that show the relation found between study in situ soil moisture profile (SMC) measurements of the Raam and Twente networks in the Netherlands, with two satellite derived (RS) vegetation indices (VIs), NIRv and VOD, during the 2018 summer. I believe that this manuscript has the quality standards of the journal and presents a very interesting work combining field measures with remote sensing measures. This is an important point. However, I have a few comments to the authors so the reader will find it easier to understand:

Thanks for writing the review and the positive feedback. Below we will respond to the comments to explain and/or elaborate.

1) In the abstract you used a lot of acronyms and that is ok. But when you say "and its effect on GPP in models" I suggest to put first what is that GPP. You use NIRv and VOD but you say that they are vegetation indexes and that is fine, but what is GPP?

We agree with point, and we have decided to use the full term (Gross Primary Productivity) instead of the abbreviation.

2) Deeper are the measures in soil are this really reflected by the VIs? or this is just a consequence of the correlation among depths of SMC. For me, it is hard to see that a measure at 80 cm depth can be reflected in RS bands. But the measures between 80 cm and 10 cm can be correlated. Can you show this correlations among depths?

It is true that the soil moisture measurements are correlated. However, we do not use remotely sensed observations of soil moisture, but use remotely sensed vegetation indices. Here, we assume that the vegetation indices reflect the state of the soil column reachable by roots, i.e. averaged over the whole root zone. Since the depth of the root zone (over which the water uptake takes place) is not known a priori and can even change over the course of a drought, we evaluate the relation with vegetation indices over different averaging depths. Hence when the vegetation indices show a decline, this matches with the available water in the soil column.

3) Are precipitation anomaly and SIF anomaly calculated in the same way that NIRv anomaly?

Yes those are all calculated in the same way. We have clarified this is the new version of the manuscript.

4) You really have three years. Calculating these anomalies means that you have the average of two years and then compare it with 2018. Is it right? Perhaps you should describe 2016 and 2017 as quite normal years, otherwise it looks too few years to consider the estimation a week anomaly.

This is correct. We have added values to compare meteorological values of 2016 and 2017, with long term means (in brackets): average temperatures were 10.4 °C (10.1 °C), yearly precipitation was 791 mm (782 mm), and year potential ET 584 mm (573 mm).

5) Figure 2. This figure is very important to understand this nice work. You should improve it as you talk about black lines (almost I cannot see it), dashed lines, etc. Please, make it more clear. I imaging that this is the average of an area. Isn't it? If I understood it right just indicate it in the label of this figure. You mention in this label Figure 3. I think that you shouldn't. Another point is that if you improve Figure 2 then the data use from Figure 2 in Figure 3 will be easier to understand.

We have clarified the captions of these figures to better represent their contents.

6) Figure 3. You mention in the label "vegetation productivity". What I can see is the relation between VIs anomalies with SWC. Between VIs and vegetation productivity which is the relation? This relation is using a time lag of 0. Did you try the relation with some time lag of 1 or 2? You mention in the introduction the lag that exist between meteorological anomalies and VIs anomalies. Exist any lag between SWC and VIs?

We have removed vegetation productivity and replaced it with vegetation indices to avoid confusion. We did not use any time lag, as this is just a scatter plot between two variables.

7) In table 1 you show the normalized critical soil moisture content in brackets. I believe that will be more interesting to see the s.e. of this estimation.

An indication of the reliability of the determined crictial soil moisture is already presented in Figure 5 using the horizontal lines, which are indicative the of the s.e.. We believe it is more interesting to see how these determined values compare to the measured soil moisture values.

8) In the abstract you said the nonlinear relation between negative soil moisture anomalies and VIs reflects that the drought was develop weeks before the first reduction in vegetation indices. Perhaps you should explore how many weeks before.

An estimate between the offset between the first reduction in soil moisture and the first reduction in vegetation indices is represented by the horizontal lines in Figure 2. This is 3 weeks when only using NIRv data, and 2 weeks for the VOD data. We have included these numbers in the results, conclusions and abstract.

Finally, this "anatomy of" expression in the title I will change it for other or just suppressed it.

We believe using this wording nicely explains how we try to understand the different dynamics of the summer drought in this region, hence the "anatomy of". Note that referee #1 noted no issues with the title, and that we have used similar titles in 2 previous HESS publications (see Brauer et al. 2011, Anatomy of extraordinary rainfall... and Geertsema et al. 2018, Anatomy of Simultaneous Flood Peaks...).

I really enjoyed your work.

Thanks for your kind words, and thanks for your time and effort in writing this constructive comment!

**List of relevant changes**

– We have shortened the introduction, and defined the abbreviations at their introduction.

– We added information on the average spacing between the soil moisture sensors in both networks.

– We added a comparison showing how our baseline years (2016–2017) compare to long term averages over the period 1990–2019.

– We quantified the duration between the reduction in soil moisture anomalies and vegetation anomalies

– We renamed the region "Twenthe" to "Twente", to be consistent with the correct naming of the region and the soil moisture network.

– We clarified captions below Figs. 2 and 3, to better describe the figures and their link.

[revised manuscript text omitted]